# Autosomal Dominant Polycystic Kidney Disease Inflammation Biomarkers in the Tolvaptan Era

**DOI:** 10.3390/ijms26031121

**Published:** 2025-01-28

**Authors:** Tânia Lapão, Rui Barata, Cristina Jorge, Carlos Flores, Joaquim Calado

**Affiliations:** 1Unidade Local de Saúde São José, Serviço de Patologia Clínica, Centro Clínico Académico de Lisboa, 1150-199 Lisboa, Portugal; ti.rocha@ulssjose.min-saude.pt (T.L.); carlos.flores@ulssjose.min-saude.pt (C.F.); 2ToxOmics, NOVA Medical School, Universidade Nova de Lisboa, 1150-082 Lisbon, Portugal; 3Unidade Local de Saúde São José, Serviço de Nefrologia, Centro Clínico Académico de Lisboa, 1069-166 Lisboa, Portugal; rui.barata@ulssjose.min-saude.pt (R.B.); cristina.jorge2@ulssjose.min-saude.pt (C.J.)

**Keywords:** ADPKD, tolvaptan, inflammation, kidney, gene

## Abstract

With the approval of tolvaptan as the first specific medicine for the treatment of rapidly progressive Autosomal Dominant Polycystic Kidney Disease (ADPKD), biomarker discovery has gained renewed interest as it is widely recognized that these will be crucial in clinical decision-making, serving as either prognostic or predictive tools. Since the marketing authorization was first issued in 2015 for ADPKD, tolvaptan has remained the sole pharmacological compound specifically targeting the disease. For ADPKD patients it is an invaluable medicine for retarding disease progression. Although the field of overall biomarker discovery and validation has been detailed in several publications, the role of inflammation remains largely overlooked in ADPKD. The current work aims to provide the reader with an updated review of inflammation biomarkers research in ADPKD, highlighting the role of urinary MCP-1 (monocyte chemoattractant protein-1) as the most promising tool.

## 1. Introduction

Autosomal Dominant Polycystic Kidney Disease (ADPKD, MIM: 173900) is a prevalent (1:1250) inherited progressive chronic kidney disease (CKD) characterized by the clonal expansion of epithelial tubular cells, leading to extensive renal cystic dysplasia, the hallmark of the disease. Molecular and cell biology research has identified the loss of functional and structural integrity of primary cilia as the underlying cause of cystogenesis and disease progression [1]. Accordingly, the genes responsible for ADPKD encode proteins that either colocalize to renal tubule epithelia primary cilia (*PKD1*, *PKD2*, and *IFT140*) or assist in their co-translational glycosylation in the endoplasmic reticulum (*GANAB*, *DNJB11*, and *ALG9*).

ADPKD accounts for 4–10% of end-stage kidney disease (ESKD) cases worldwide [2]. With the emergence of *novel* and specific reno-protective therapies, prognostic assessment during the early stages of ADPKD has become increasingly important [3]. Several markers are associated with kidney outcomes and have even been incorporated into clinical algorithms intended to predict renal insufficiency and ESKD, such as the Mayo Imaging Classification (MIC) and Predicting Renal Outcomes in ADPKD score [4]. In addition to age, sex, hypertension, and urological events before the age of 35 years, the markers with the highest sensitivity and specificity are imaging and genotypic information: elevated MIC scores and truncating *PKD1* mutations are predictive of ESKD and estimated glomerular filtration rate (eGFR) endpoints, with genotype enhancing imaging predictions and imaging predictions, in turn, improving genotype assessments [5]. The gradual expansion of renal cysts results in bilateral kidney enlargement, measured as an increase in total kidney volume (TKV), preceding renal insufficiency. Individuals with the highest MIC score (1E) display a median age for ESKD of 45.1 years versus 71.2 years for MIC 1B [5]. Serial measurements of the height-adjusted TKV (htTKV) using Magnetic Resonance Imaging (MRI) or Computerized Tomography (CT) scans are recommended for proper assessment of ADPKD risk progression [3], but the associated costs (MRI) and radiation exposure (CT) are significant caveats while using MIC as a risk prediction tool.

Regarding genotype-phenotype correlations, several studies have linked *PKD1* truncating mutations to worse outcomes. Patients harboring *PKD1* truncating mutations reach ESKD at a median age of 55.6 years versus 67.9 years for those carrying non-truncating variants [6]. However, because of the high cost, genetic testing is not routinely recommended, and for most patients, genotype information is missing. These are clinically validated biomarkers, but given the limitations mentioned above, there is a great demand for affordable, non-invasive biomarkers to assist clinicians in accurately predicting renal outcomes in ADPKD and selecting patients for disease-specific modifying medicines.

Tolvaptan (Jinarc, Otsuka Pharmaceuticals) is the first and only FDA/EMA-approved treatment indicated to slow kidney function decline in adults at risk of rapidly progressing ADPKD. The randomized controlled trial TEMPO 3:4 [7] provided the primary evidence for the drug’s approval. The TEMPO 3:4 primary endpoint was the rate of TKV change from baseline relative to placebo, while the composite secondary endpoint was the time to several ADPKD outcomes, including worsening renal function. Tolvaptan significantly reduced both endpoints for this ADPKD study population with well-preserved renal function at baseline (eGFR > 60 mL/min/1.73 m^2^). The REPRISE trial [8] extended the benefits of tolvaptan to a cohort with more advanced CKD stages (eGFR 25–65 mL/min/1.73 m^2^).

As benefits are restricted to patients with rapidly progressing disease (i.e., ESKD developing before 55–60 years of age), identifying patients at risk of rapid progression early in life, even before GFR begins to decline, is paramount in this tolvaptan post-approval era. Based on the TEMPO 3:4 and REPRISE trials, early initiation of tolvaptan (eGFR 90 mL/min/1.73 m^2^) may delay ESKD by up to 7.3 years, compared with only 2.3 years for late initiation (eGFR 30 mL/min/1.73 m^2^) [9].

## 2. Tolvaptan Structure

Arginine vasopressin (AVP), also known as the antidiuretic hormone, is a neuropeptide produced by the hypothalamus and systemically released from the posterior pituitary. Its actions are mediated by the V_1a_, V_1b_, and V_2_ receptors that account for several physiological effects, including vasoconstriction, platelet aggregation, hemostasis, and regulation of extracellular osmolality [10]. In particular, AVP binding to the V_2_ receptor in the basolateral surface of epithelial principal cells of the kidney’s collecting duct induces reabsorption of water. The V_2_ receptor is a G protein-coupled receptor that generates intracellular cyclic AMP (cAMP) culminating in the trafficking of aquaporin-2 to the apical surface, increasing membrane permeability to water.

Tolvaptan (C_26_H_25_ClN_2_O_3_, CAS identifier 150683-30-0) is an orally active, nonpeptide arginine vasopressin V_2_- receptor selective antagonist [11]. The chemical structure of this benzazepine derivative is shown in Figure 1.

Early in vitro studies showed that in HeLa cells transfected with V_2_, V_1a_, or V_1b_ human receptor constructs, tolvaptan antagonized AVP binding to the human V_2_ receptor 29 times more selectively than to V_1a_ (Ki of 0.43 versus 12.3 nM) and there was no inhibition observed for V_1b_. Pharmacodynamic profiling in rats revealed a dose-dependent aquaresis and in animal models of hyponatremia tolvaptan corrected plasma sodium levels [11].

Clinical trials with tolvaptan in heart failure where hyponatremia is a prevalent finding did not show any impact on long-term mortality or heart failure-related morbidity [12]. These studies, however, demonstrated hyponatremia reversal. Accordingly, the FDA first approved Tolvaptan (Samsca, Otsuka Pharmaceuticals) for individuals with significant hypervolemic and euvolemic hyponatremia in 2009.

Simultaneously, preclinical research in polycystic disease with tolvaptan was conducted with the aim of testing whether reverting the increased intracellular cAMP, a finding common to different cystic disease models, could attenuate disease progression. The mechanisms by which cAMP promotes cyst growth are both through inducing water and chloride secretion mediated by the apical chloride channel (CFTR) insertion and also by inducing epithelial proliferation through the ERK/MAPK pathways [13]. Preliminary rodent models, including the *pcy* mouse [14], revealed a beneficial effect of tolvaptan in attenuating disease progression. These observations impelled tolvaptan into clinical research and the randomized clinical trials that ultimately led to tolvaptan’s approval for ADPKD in 2015 by the EMA, and in 2018 by the FDA (Jinarc, Otsuka Pharmaceuticals).

Recently, the structural basis for the binding of tolvaptan to the V2 receptor has been solved [15]. Upon binding to extracellular loops 2–3, the molecule stabilizes in a deeper binding pocket. The G-protein coupled V_2_ receptor residues R181, Y205, F287, and F178 were found to be crucial for this binding.

## 3. Biomarkers Research in ADPKD

The search for biomarkers in ADPKD long preceded the introduction of tolvaptan into clinical practice. According to the FDA/NIH BEST (FDA-NIH Biomarker Working Group. BEST) [16], a biomarker is defined as a “characteristic that is measured as an indicator of normal biological processes, pathogenic processes, or biological responses to an exposure or intervention.” In ADPKD, research interest has revolved around prognostic and, when medical intervention is considered, predictive, monitoring, or response biomarkers [17].

The rationale behind tolvaptan’s use in ADPKD is based on the upregulation of the vasopressin V_2_ receptor that drives increased cAMP production, a key molecular hallmark of the cystic phenotype [18]. These findings parallel observations of urinary concentration defects and increased plasma levels of AVP in animal models and ADPKD patients. Accordingly, the C-terminal part of the AVP precursor copeptin, which is stable in plasma and more practical than AVP in clinical settings [19], has been evaluated in ADPKD. Following water privation, ADPKD patients with preserved eGFR (>60 mL/min/1.73 m^2^) were found to have lower maximal urine-concentrating capacity and higher AVP and copeptin concentrations compared with controls in a study of 30 patients [20]. In a cross-sectional observational study of 102 ADPKD patients, copeptin levels were associated with lower eGFR, effective renal blood flow, and higher TKV [21]. Additionally, in a follow-up study of 79 ADPKD patients, copeptin was associated with changes in inulin clearance and eGFR in the short- and long-term follow-up, respectively [22]. Furthermore, research that included 52 ADPKD patients found that higher copeptin levels correlated with worse kidney outcomes (assessed by eGFR and TKV), while apelin, released in response to low plasma osmolality, inversely correlated with copeptin [22]. These studies validated plasma copeptin as a surrogate marker of disease progression, though it has yet to be incorporated into clinical decision algorithms [4].

Urinary-concentrating biomarkers are particularly appealing in ADPKD, either as prognostic or monitoring biomarkers. In the TEMPO 3:4 cohort, baseline urine osmolality (Uosm) negatively correlated with age and TKV, while positively correlated with eGFR [23]. The greatest renal benefit occurred in subjects with greater Uosm suppression, suggesting that Uosm could be an excellent monitoring/response biomarker [23]. Data from the Developing Interventions to Halt Progression of Autosomal Dominant Polycystic Kidney Disease (DIPAK)-1 observational cohort study revealed that in 583 patients with a mean follow-up of four years, the urine-to-plasma urea ratio (UPureaR), a better measure of urine concentration defect in early ADPKD, was associated with the rate of eGFR decline [24].

Although research in biomarkers reflecting the role of kidney concentrating defects in the pathogenesis of ADPKD has been fostered by the introduction of tolvaptan in nephrology practice, many other diverse blood and urinary biomarkers have been evaluated in ADPKD. These range from traditional glomerular injury markers (e.g., urinary excretion of albumin and IgG) and proximal tubular markers [e.g., β2-microglobulin (β2MG), Kidney Injury Molecule-1 (KIM-1), Liver-type Fatty Acid-Binding Protein (L-FABP), N-acetyl-β-D-glucosaminidase (NAG), and Neutrophil Gelatinase-Associated Lipocalin (NGAL)] to distal tubular markers [e.g., Heart-type FABP] and state-of-the-art urinary metabolomics and exosomal miRNAs [17,25,26,27]. However, none of the biochemical markers examined have been shown to perform better than current imaging biomarkers, although several significantly improve prognostication when added to TKV.

Essentially a developmental kidney disorder, ADPKD can, innovatively, be regarded as a tubular disease with an inflammatory component, and several studies have used this hypothesis to promote biomarker discovery. Regarding proximal tubule injury-related markers, KIM-1, β2MG, and NGAL are the most promising [27]. In the following sections, we will review the available data concerning inflammation biomarkers in ADPKD and its relation to cyst growth and disease progression.

## 4. Cyst Initiation and Cyst Growth in ADPKD

ADPKD etiopathogenesis derives from two distinct and sequential processes: (i) an initiating event in which the *PKD1/PKD2* functional activity falls below a critical threshold, triggering a cascade of signaling events; and (ii) cyst growth mediated by cellular proliferation, fluid secretion, and extracellular matrix deposition.

Normal tubule structure and integrity require a critical level of cellular functional activities of Polycystins 1 and/or 2 (PC1, PC2), the proteins encoded by *PKD1* and *PKD2* genes [28]. According to the genetic mechanism proposed in ADPKD of a “two hit” event, by which there is a somatic inactivation of the remaining wildtype allele, cellular *PKD1/PKD2* functional activity may fall below a critical threshold, and cyst initiation ensues. Alternatively, in the advent of biallelic disease as seen in the rare cases of homozygous hypomorph germline mutations [29], the “second hit” is not necessary for cyst initiation since the total functional level of the two Polycystins may be lower than the threshold, depending on the metabolic state of the epithelial cell.

Cyst initiation is primarily a molecular genetics process. In contrast, cyst growth derives from continuous cell proliferation, fluid secretion, and aberrant extracellular matrix composition [30], with inflammation and fibrosis playing a major role. Cyst growth occurs due to the activation of multiple cellular and metabolic pathways once the functional activity threshold is crossed.

Figure 2 illustrates the process. During the early genetically driven event of cyst initiation, failure of PC1/PC2 to reach the sustainable biological threshold initiates cystic phenotypic changes in the epithelial cell. Further cyst progression involves activation of multiple pathways, leading to increased epithelial tubular proliferation, CFTR-mediated fluid secretion of water and chloride, and abnormal matrix deposition with quintessential myofibroblast and macrophage infiltration.

Surprisingly, the phenotype changes observed once polycystic genes are inactivated are not irreversible. In a mouse model permissive to the inactivation of *Pkd1* or *Pkd2* as well as subsequent gene reactivation, it was discovered that cyst formation is reversible, with cystic structures returning to normal nephrons. There was also an increase in autophagic flux, decreases in tubular cell proliferation, extracellular matrix deposition, and myofibroblast activation, and, relevant to the present review, overall kidney inflammation [31].

## 5. Injury-Repair Mechanism in ADPKD

Following an insult, the kidneys are able to repair the injury by inducing the proliferation of the surviving tubular epithelia. Inflammation is an important component of the injury-repair mechanism, as ischemic injury prompts tubular cells to secrete pro-inflammatory and chemotactic cytokines such as tumor necrosis factor-α (TNF-α), monocyte chemoattractant protein-1, also known as CC-chemokine ligand 2 (MCP-1/CCL2), IL-8, IL-6, or IL-1β. These cytokines recruit and activate inflammatory cells [32]. With previously healthy tubular epithelia and in cases of limited damage, repair is partially dependent on the switch from the M1 (pro-inflammatory classically activated) to M2-like (healing repair) macrophage phenotypes (see below) that help in the resolution of inflammation and in restoring tissue integrity. If, on the other hand, the damage is too extensive or the insult persists, proliferating cells may arrest in the G2/M phase, the M1 to M2 switch will not take place, and a state of chronic inflammation ensues with progressive collagen deposition and loss of the normal parenchymal structure, leading to CKD [33].

In ADPKD, low PC1/PC2 activity can spontaneously induce epithelial proliferation. This process is exacerbated further if superimposed injury occurs, which, together with mechanical stress on adjacent tissue and vessels, pro-inflammatory cytokine secretion, and activation of metabolic pathways, promotes cyst progression [28,33]. The early pro-inflammatory response activates NF-κB and JAK-STAT pathways, resulting in further transcription of pro-inflammatory cytokines [34]. In an attempt to counter-balance inflammation, incoming leucocytes release pro-fibrotic factors that, together with growth factors such as transforming growth factor and epidermal growth factors, will promote fibrosis, epithelial–mesenchymal transformation, activation of myofibroblasts, and extracellular matrix deposition [33].

Recent work identified the primary cilia as a key regulator of the injury-repair process, by which their disruption traps epithelial cells in a persistent injury state, signaling maladaptive repair responses to nearby resident macrophages [35]. Inflammation and immune pathways are crucial for this maladaptive repair response and are reviewed in the following section.

## 6. Inflammation and Immune Pathway Biomarkers

Early studies have documented kidney interstitial inflammation and fibrosis in ADPKD, with pro-inflammatory molecules, namely TNF-α and MCP-1/CCL2, being present in high concentrations in the urine and cyst fluids. Infiltrating CD68^+^ macrophages, together with CD3^+^ lymphocytes, were also reported in seminal renal histology studies [36]. In advanced stages of the disease, genes coding for alternative macrophage activation pathways (M2-like) were found to be the most upregulated. As reviewed in the previous section, the injury-repair mechanism is central to ADPKD progression, with injury promoting cyst formation and cyst enlargement, which perpetuates local injury and inflammation.

While the roles of innate immunity, macrophages, and injury repair are well established in ADPKD etiopathogenesis, less is known about adaptative immunity and lymphocytes or their usefulness as biomarkers.

### 6.1. Innate Immunity and Inflammation

Earlier studies have identified infiltrating macrophages in polycystic kidneys of ADPKD individuals and in several rodent models of the disease. Macrophages are swiftly generated from monocytes following the appropriate stimuli. They are critical immune effector cells distinguished by their exceptional phagocytic ability. Depending on the interactions between macrophages and specific immune cell subsets, macrophages can undergo a full spectrum of activation, ranging from an M1-like (classically activated) to an M2-like (alternatively activated) phenotype [37]. Both can be induced through either adaptative (T helper) or innate immune responses. Classically activated macrophages arise in response to interferon-γ (IFN-γ), produced by T helper 1 (T_H_1) and natural killer cells or by antigen-presenting cells derived TNF-α. The M2-like phenotype is induced by IL-4 either originating from T_H_2 cells or specifically from granulocytes (innate immune response).

Through different signaling pathways, these macrophages have different roles in immune, inflammatory, and injury-repair responses: M1-like are pro-inflammatory effector macrophages that, in ADPKD, inflict direct damage to renal tubules, while M2-like, traditionally regarded as injury/inflammation repair macrophages, can, in ADPKD, further stimulate tubular proliferation and cyst growth instead of promoting healing (maladaptive repair) [33,37].

MCP-1/CCL2, a potent chemotactic factor for monocytes, is produced by a large variety of cell types [38]. In addition to epithelial cells, monocytes/macrophages are themselves a major source of MCP-1/CCL2, therefore establishing a positive feedback loop at sites of injury and inflammation (Figure 2). MCP-1/CCL2 effects are mediated by binding to the CCR2 receptor expressed in endothelial and mononuclear cells. In these cells, through specific signaling pathways, CCR2 activation will ultimately attract leucocytes to sites of inflammation and injury. MCP-1/CCL2 also influences adaptative immunity, both by inducing T_H_2 differentiation and IL-4 secretion [38], resulting in the M1 to M2-like phenotype macrophage switch that further drives tubular proliferation and cyst growth in ADPKD.

#### 6.1.1. MCP-1/CC2L in Murine Models of Polycystic Kidney Disease

The role of MCP-1/CCL2 and infiltrating macrophages in ADPKD has been addressed in several murine models, both orthologous and nonorthologous, of human polycystic disease. Increased expression of *Mcp1*, accompanied by accumulation of macrophages, was reported in the kidneys of Han:SPRD polycystic rats, a nonorthologous model of ADPKD [39]. Also, microarray analysis of kidneys from cystic *cpk* mice, a nonorthologous model of autosomal recessive polycystic, demonstrated that genes involved in the innate immune response were the most significantly upregulated [40]. In orthologous models of *PKD1* and *PKD2*, it was found that macrophages infiltrated the cystic kidneys 10 times more compared to controls and that by depleting macrophages in these mice, the cystic phenotype was attenuated [41]. Also, cultured Pc1 null kidney cells were found to secrete increased amounts of Mcp1 and Cxcl16. The researchers postulated that these could signal the accumulation of alternatively activated macrophages in cystic areas which, in turn, would promote cyst growth via secretion of macrophage-derived factors that stimulate adjacent tubular cell proliferation [41].

In line with the above-mentioned research, in conditional single (*Pkd1*) and double (*Pkd1* and *Mcp1*) knockout (SKO and DKO) renal tubule-specific models, it was confirmed that the tubular epithelial cell is the primary source of Mcp1 after the loss of Pc1 and that the former recruits circulating monocytes [42]. In the DKO and with pharmacological antagonism of the Mcp1 receptor Ccr2 in the SKO, decreased macrophage numbers, cyst growth, and serum creatinine were observed. In the initial phase of injury, macrophages exhibit a proinflammatory profile promoting cyst expansion in a proliferation-independent manner, while in later phases, and upon alternative activation (M1 to M2-like switch), they directly drive cell proliferation [42]. Concerning the benefits of Mcp1 antagonism, the findings are contradictory. While the above-mentioned Ccr2 receptor antagonism was found to positively influence cyst growth and mice mortality, in the PCK rat, a nonorthologous model, the use of the Mcp1 inhibitor bindarit showed no benefit in cyst growth, although it limited interstitial inflammation [43]. The reason for this discrepancy may reside in the fact that bindarit failed to completely abrogate macrophage accumulation in the renal interstitium.

Cytokines and chemokines (chemoattractant cytokines) are pleiotropic secreted proteins that regulate immune cell growth, differentiation, activation, and trafficking. They play a central role in coordinating inflammation and repair by recruiting and activating specific cell populations. As mentioned, several studies have documented the pivotal role of cytokines in cystogenesis and disease progression, as exemplified by the increased levels of IL-1β and TNF-α by ELISA, or osteopontin with mass spectrometry [44,45] within the cyst. TNF-α is the main inducer of MCP-1/CCL2 expression, although other pro-inflammatory cytokines like IL-1, IL-4, or IFN-γ may have a comparable role [38,46]. This finding highlights the importance of the TNF-α/MCP-1/CCL2 axis in inflammation, cyst formation, and disease progression in ADPKD, a role that has been corroborated in rodent models where exposure to TNF-α increased the incidence of cyst formation [33].

The macrophage migration inhibitory factor (MIF), another pro-inflammatory cytokine with an important role in the recruitment of innate and adaptive immune cells to sites of inflammation, was also found to drive TNF-α and MCP-1/CCL2 expression in Pc1-deficient murine kidneys, establishing a paracrine feedback loop by which the cystic fluid secreted TNF-α induces MIF expression in renal epithelial cells [47]. The authors postulated a role for MIF in cystogenesis through both macrophage-dependent (secondary effect) and macrophage-independent (primary effect) cystogenesis, with the latter mediated by an upstream influence on ERK, AMPK, mTOR, Rb/E2F, and p53 signaling pathways [47].

As recently reported for a mouse model enabling both inactivation of *Pkd1* or *Pkd2* and subsequent gene reactivation, it was discovered that cyst formation is indeed (and somehow surprisingly) reversible [31]. In particular, the observed increases in inflammation, extracellular matrix deposition, and myofibroblast activation following gene inactivation were reversed once the polycystic genes were reactivated. Specifically, at the transcriptional level, the observed increases of mRNA *Mcp1* and F4/80 (a marker for murine macrophages) were reversed upon gene reactivation. These changes were paralleled by the number of infiltrating inflammatory cells, including F4/80 macrophages, and the abundance of TNF-α at pericystic localization [31]. The authors concluded that the swift reversal of these biochemical and cellular changes indicates a causal link between polycystin function and the immediate triggers for activating and deactivating inflammatory responses in ADPKD kidneys.

Altogether, these investigations have postulated a role for urinary and/or circulating cytokines, chemokines in particular, as prognostic or monitoring biomarkers in ADPKD. Accordingly, some of them have been assessed in human ADPKD cohorts.

#### 6.1.2. Urinary Cytokines as Biomarkers in ADPKD

In a preliminary study with 55 patients, MCP-1/CCL2 urinary excretion was reported to be elevated in ADPKD individuals compared to controls [48]. No significant differences were observed either in serum levels or in the apical medium as a measure of secretion by cystic epithelia when compared to normal serum controls and human kidney cortex cells, respectively. Still, MCP-1/CCL2 levels in the cystic fluid were much greater than in serum, arguing for an active MCP-1/CCL2 secretion by cystic epithelia. In a cohort of 102 ADPKD patients with ADPKD, β2MG, NGAL, and H-FAP were associated with effective renal blood flow and measured GFR, while MCP-1/CCL2, KIM-1, and NGAL related to TKV [27]. Additional studies have since consistently found that urinary MCP-1/CCL2 is an important risk marker for ADPKD progression (Table 1). In an observational longitudinal study that included 104 patients, with a mean follow-up of 3.8 years, urine samples were tested for several biomarkers, targeting glomerular, tubular, and inflammatory injury. Using commercially available assays, the authors found that urinary MCP-1/CC2L and β2MG excretion were strongly associated with eGFR decline in ADPKD and offered added predictive value beyond traditional risk markers [49]. These observations were replicated in a follow-up study of 152 patients under standardized care, where β2MG and MCP-1/CC2L again showed the strongest association with rapidly progressive disease [50]. These findings highlight the role of not only inflammation but, surprisingly, of proximal tubular damage in ADPKD progression, a disease for which cysts are predicted to originate mostly in the more distal tubule. Notably, in the TEMPO3:4 trial, tolvaptan reduced urinary MCP-1/CCL2 excretion compared to placebo across CKD stages 1–3. The researchers hypothesized that tolvaptan inhibited the proliferation of cells with a paracrine secretion of MCP-1/CCL2 [51]. A recent cross-sectional study involving 233 ADPKD patients from the German AD(H)PKD registry evaluated serum cytokines profile and assessed their utility as biomarkers for disease severity and progression [52]. Five cytokines exhibited significant differences between patients and controls: IL-6, IL-8, MCP-1/CCL2, TNF-α, and IFN-γ. However, regarding the relationship with eGFR, only TNF-α demonstrated a consistent negative correlation [52]. These observations reinforce, once again, the centrality of TNF-α as an upstream effector and highlight the consequential role of MCP-1/CCL2 in disease progression.

In other inflammatory kidney diseases such as ANCA-associated vasculitis, an autoimmune glomerulonephritis, the urinary excretion of soluble CD163 (usCD163), a scavenger receptor shed by activated M2-like macrophages, was highly specific and sensitive for relapsing disease [53]. usCD163 is entering nephrology clinical practice as a disease activity biomarker; however, its role in ADPKD has not been addressed.

### 6.2. Adaptative Immunity and Inflammation

The role of adaptative immunity in ADPKD has been far less studied than that of its innate counterpart. Early studies in human ADPKD specimens have documented scarce interstitial infiltrates in early and ESKD renal failure specimens, with a predominance of CD4^+^ lymphocytes as assessed by immunohistochemistry (IHC) [36]. Lymphocytic infiltrating cells were also described in cystic kidneys of rodent models [54,55] and more recently T-cell subpopulations were characterized in the C57Bl/6 *Pkd1*^RC/RC^ ADPKD orthologous model [56]. Using flow cytometry of renal single-cell suspensions and IHC of kidney specimens, the authors reported an increase in T-cell numbers, of both the CD8 and CD4 phenotypes, which correlated with disease severity [56]. T-cell activation was most prominent among the CD8^+^ phenotype, the depletion of which led to an increase in the disease burden, portraying an inhibiting effect in cystogenesis for CD8^+^ T-cells. These findings highlight the changes in adaptive immune cells in ADPKD and the potential usefulness of urinary T-cells as biomarkers in ADPKD.

In addition, the landscape of infiltrating T-cells in the kidney tissue of ADPKD patients was studied by flow cytometry [57]. In six patients, flow cytometry revealed increased numbers of intrarenal CD3^+^ cells of both helper and cytotoxic phenotypes compared to controls. Most interestingly, the authors discovered in 30 ADPKD patients an inverse correlation between CD4^+^ urinary index and eGFR or eGFR decline, pointing to urine T-cells as a candidate marker for disease activity in ADPKD [57].

**Table 1 ijms-26-01121-t001:** ADPKD inflammation biomarkers studies in humans.

Study (Reference)	Patients (n)	Research Method	Outcomes	Results
Zheng et al. [48]	55	Observational/cross-sectional	MCP-1/CCL2 in urine, cyst fluid and serum MCP-1/CCL2 production by cultured cystic mural cells	MCP-1/CCL2 urinary excretion increased in ADPKD compared to controls
Meijer et al. [27]	102	Observational/cross-sectional	IgG, KIM-1, NAG, NGAL, β2MG, H-FABP and MCP-1/CCL2 in urine	MCP-1/CCL2 (and NGAL, β2MG, and H-FABP) urinary excretion associated with TKV
Grantham et al. [51]	1307	Interventional	MCP-1/CCL2 in urine	MCP-1/CCL2 urinary excretion decreased with tolvaptan
Messchendorp et al. [49]	104	Observational/follow-up	albumin, IgG, KIM-1, NAG, β2MG, H-FABP, MIF, NGAL and MCP-1/CCL2 in urine	MCP-1/CCL2 (and β2MG) urinary excretion correlated with GFR decline
Messchendorp et al. [50]	152	Observational/follow-up	albumin, IgG, KIM-1, β2MG, H-FABP, NGAL and MCP-1/CCL2 in urine	MCP-1/CCL2 (and β2MG) urinary excretion correlated with GFR decline
Arjune et al. [52]	233	Observational/cross-sectional	IL-1β, IL-2, IL-6, IL-8, IL-10, IL-13, IFN-γ, MCP-1/CCL2 and TNF-α in serum	IL-6, IL-8, MCP-1, TNF-α and IFN-γ is serum increased in ADPKD compared to controls
Zimmerman et al. [57]	30	Observational/cross-sectional	T-cell populations in urine and kidney tissue	CD4+ index in the urine correlated with GFR decline

## 7. Conclusions

Inflammation in ADPKD has only recently been evaluated either in the disease’s pathogenesis and its usefulness in selecting therapeutic targets and outcome/prognostic biomarkers. Inflammation may not play a part in cyst initiation but it secondarily promotes disease progression through macrophage recruitment that, upon alternative activation (the M1 to M2-like switch), perpetuates a maladaptive injury-repair response that ultimately results in cyst inflation and tubular epithelial proliferation and increased extracellular matrix deposition. The inflammatory pathway has been selected as a therapeutic target, as exemplified by a phase 3 clinical trial using bardoxolone, an Nrf2-Keap1 activator with anti-inflammatory properties (ClinicalTrials.gov ID NCT03918447). Finally, and as reviewed, urinary MCP-1/CCL2 is emerging as a biomarker that adds relevant prognostic information to MIC and genotype information. The role of other promising innate inflammatory biomarkers, like usCD163, remains to be clarified. Less is known regarding the adaptative immune response in ADPKD.

## Figures and Tables

**Figure 1 ijms-26-01121-f001:**
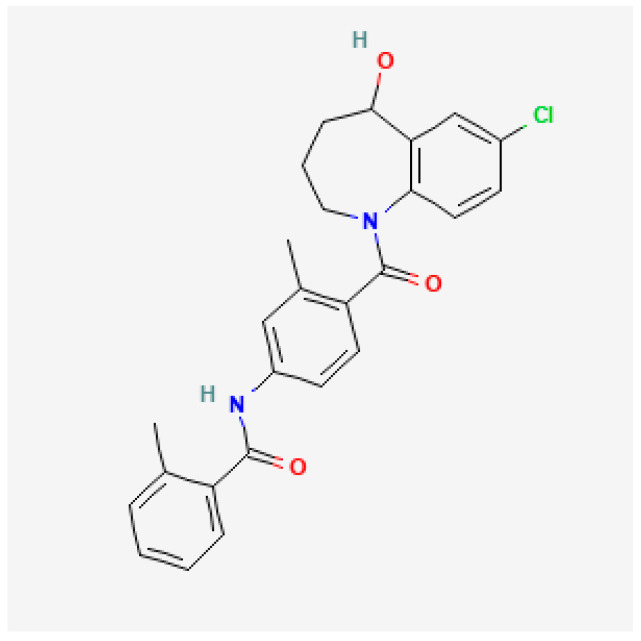
Chemical structure of tolvaptan (https://pubchem.ncbi.nlm.nih.gov/compound/216237, accessed on 16 January 2025).

**Figure 2 ijms-26-01121-f002:**
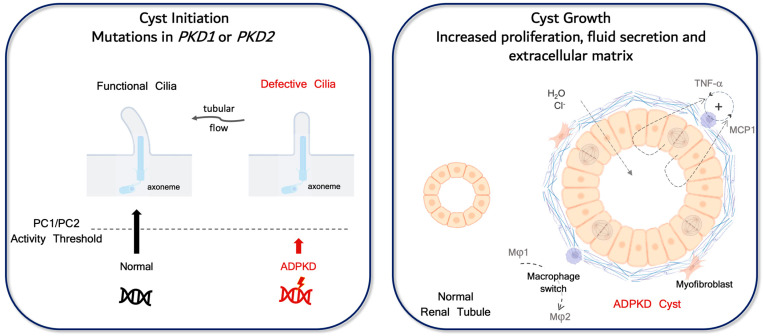
Cyst initiation and cyst growth. In the left panel, we illustrate the PC1/PC2 functional activity threshold necessary to maintain the structural integrity of the epithelial primary cilia. Cells harboring germline mutations in cystic genes coupled with somatic inactivation of the remaining gene copy (“second-hit”) will fail to reach the threshold. Defective cilia will no longer sense the luminal flow and epithelial cells will undergo dedifferentiation, paving the way for cyst initiation. Cyst growth, on the other hand, is portrayed in the right panel. It relies on the activation of multiple pathways, all leading to an increase in tubular epithelia proliferation, fluid secretion of water and chloride, and abnormal matrix deposition. Central for disease progression is the maladaptive repair response mediated by macrophages. The paracrine secretion of either TNF-α and MCP-1/CCL2 (MCP1) by cystic epithelia generates a positive feedback loop that will recruit more monocytes and macrophages to the pericystic interstitium. The continuous inflow of M1-like macrophages perpetuates inflammation and injury, while the switch to an M2-like phenotype (alternative activation) will lead to maladaptive repair instead of healing repair, with further epithelial proliferation and cyst growth.

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
