# Peer review of "Autosomal Dominant Polycystic Kidney Disease Inflammation Biomarkers in the Tolvaptan Era"

_ijms, 2025, doi:10.3390/ijms26031121_

Round 1

Reviewer 1 Report

Comments and Suggestions for Authors

I read with interest this review manuscript by Lapao et al. looking at the current state of inflammatory biomarker utility within ADPKD, with special emphasis on MCP-1.  I thought the review was interesting, informative, and well-written.  It makes a valuable contribution to the literature, and I expect will be of interest to readers in the field.  

My biggest suggestion would be a revision of Table 1.  The font is large and could be decreased.  The Research Methods column is too wide, and more information should be given in outcomes and results.  These headings also seem somewhat redundant.  More information on number of patients and other details of the study would be helpful.  The authors could also make more space by utilizing the Reference section to give author information.  More space could also be made by removing the immunity column, which didn't add much additional information.    

Author Response

Sir,

Thank you so much for your positive feedback as well the suggestions regarding table of the manuscript. We believe that your contributions will definite make this review more worthful.

Reply to specific comments

Comment 1. The font is large and could be decreased: we have change it accordingly;

Comment 2. The Research Methods column is too wide, and more information should be given in outcomes and results. More information on number of patients and other details of the study would be helpful: we have included the number of patients;

Comment 3. The authors could also make more space by utilizing the Reference section to give author information: we have proceeded accordingly

Comment 4: More space could also be made by removing the immunity column, which didn't add much additional information: we couldn’t agree more and as such we have remove it.

Reviewer 2 Report

Comments and Suggestions for Authors

Comments on the review article by Lapão et al. submitted to JIMS mdpi titled: ADPKD Inflammation Biomarkers in the Tolvaptan Era

This is a very interesting work that primarily focuses on the role of inflammation biomarkers in Autosomal Dominant Polycystic Kidney Disease (ADPKD) and their potential utility in assessing disease severity and progression. It emphasizes the importance of identifying reliable biomarkers that can aid in clinical decision-making and improve patient outcomes, particularly in the context of treatment with tolvaptan

Tolvaptan is a medication that is primarily used to treat Autosomal Dominant Polycystic Kidney Disease (ADPKD) and certain conditions related to heart failure. It is a selective vasopressin V2 receptor antagonist, which means it blocks the action of vasopressin (also known as antidiuretic hormone) at the V2 receptors in the kidneys. This action leads to increased excretion of free water, thereby reducing urine concentration and helping to manage fluid balance.

Genarally it is a very interesting work, however, some parts of it should be improved and some significant information added before publication. My comments as below:

The use of the term ‘tolvaptan era’ is justified in the context of the treatment of Autosomal Dominant Cystic Kidney Disease (ADPKD). The term refers to the period when tolvaptan became the first specific drug approved for the treatment of this disease, which had a significant impact on the approach to the management of ADPKD.

As a review, the manuscript is quite short. In my opinion, it lacks many important details. First of all, it would be beneficial to mention the molecular structure of tolvaptan.

Regarding the chemical formula and the detailed mechanism of action, the document does not provide the specific chemical formula for tolvaptan. However, it does explain its mechanism of action in the context of ADPKD.

Unfortunately, the document does not include specific details about the chemical structure of tolvaptan. The text primarily focuses on the clinical applications of tolvaptan in treating Autosomal Dominant Polycystic Kidney Disease (ADPKD), its mechanism of action, and associated biomarkers, rather than delving into its chemical structure or composition. The review mentions that tolvaptan works by inhibiting the upregulation of the vasopressin V2 receptor, which is linked to increased cyclic AMP (cAMP) production—a key factor in the cystic phenotype of ADPKD. By blocking this receptor, tolvaptan helps mitigate the effects of vasopressin, which contribute to cyst growth and the decline of kidney function in ADPKD patients.

Adding references to studies that explore the interactions of specific chemical groups in tolvaptan with elements of the molecular organization of human cells would greatly strengthen the manuscript.

Please rewrite the Abstract to indicate the irreplaceable effect of tolvaptan in ADPKP treatment.

Other remarks:

Note the high level of similarity of the manuscript to other scientific papers according to Trintenticate as 42%. This is an unacceptable level. Rewrite the relevant parts of the manuscript so that the degree of similarity does not exceed 25%.

Please pay attention to the article formatting:

1.      Unnecessary full stop at the end of Keywords

2.      Formatting of Table 1

3.      Formatting of references and citations within the manuscript text.

4.      Numbering of pages

Author Response

Sir,

Thank you so much for your feedback and assertive remarks on the manuscript recension. We believe that your contributions will definitely make this review more valuable.

We have submitted a revised mnsc that, we feel, will address the reviewer’s  concerns. Revised parts of the manuscript are written in red.

#Reply to specific major comments

Comment 1: As a review, the manuscript is quite short. In my opinion, it lacks many important details. First of all, it would be beneficial to mention the molecular structure of tolvaptan. Regarding the chemical formula and the detailed mechanism of action, the document does not provide the specific chemical formula for tolvaptan. However, it does explain its mechanism of action in the context of ADPKD. Unfortunately, the document does not include specific details about the chemical structure of tolvaptan. The text primarily focuses on the clinical applications of tolvaptan in treating Autosomal Dominant Polycystic Kidney Disease (ADPKD), its mechanism of action, and associated biomarkers, rather than delving into its chemical structure or composition. Adding references to studies that explore the interactions of specific chemical groups in tolvaptan with elements of the molecular organization of human cells would greatly strengthen the manuscript.

Reply: We have added a new section entitled “Tolvaptan Structure”, detailing not only on the chemical structure of tolvaptan (figure; pubchem; CAS) but also its interaction with the natural occurring ligand, vasopressin (AVP). Details on the tolvaptan in-vitro inhibitory kinetic studies are provided, as well preliminary pre-clinical research and the most relevant clinical trials regarding the first indication for which tolvaptan was granted FDA approval; reversal of hyponatremia in the heart failure setting. Finally, the residues necessary for the interaction of tolvaptan with the AVP V2 receptor have been recently structurally solved at a molecular level and these are mentioned and referenced. Six new references were accrued that provide the body of evidence for this section.

Comment 2: Please rewrite the Abstract to indicate the irreplaceable effect of tolvaptan in ADPKP treatment.

Reply: This has been addressed by including additional sentences to the abstract (lines 14-16)

#Other remarks:

Comment i): Note the high level of similarity of the manuscript to other scientific papers according to Trintenticate as 42%. This is an unacceptable level. Rewrite the relevant parts of the manuscript so that the degree of similarity does not exceed 25%.

Reply: We have rewritten sentences/paragraphs (and parts) that deemed to be similar to the original published work that substantiate this review, while trying to remain faithful to the authors’ statements. We have rerun the text with the on-line software Duplichecker and thought to have minimize the risk of plagiarism imputation.

Comment ii): unnecessary full stop at the end of Keywords.

Reply: Full stop removed

Comment iii): Formatting of Table 1

Reply: Properly formatted, this was also a major issue of another reviewer.

Comment iv): Formatting of references and citations within the manuscript text.

Reply: Brackets are now used for referencing within the text.

Comment v) Numbering of pages

Reply: They appear on right heading of each page.